# Design and Modeling of a Fully Integrated Microring-Based Photonic Sensing System for Liquid Refractometry

**DOI:** 10.3390/s22239553

**Published:** 2022-12-06

**Authors:** Grigory Voronkov, Aida Zakoyan, Vladislav Ivanov, Dmitry Iraev, Ivan Stepanov, Roman Yuldashev, Elizaveta Grakhova, Vladimir Lyubopytov, Oleg Morozov, Ruslan Kutluyarov

**Affiliations:** 1Ufa University of Science and Technology, 32, Z. Validi St., Ufa 450076, Russia; 2Kazan National Research Technical University named after A. N. Tupolev-KAI (KNRTU-KAI), 10, Karl Marx Street, Kazan 420111, Russia; 3Kazan Federal University, 18, Kremlyovskaya Str., Kazan 420008, Russia

**Keywords:** integrated photonics, silicon photonics, refractometry, optoelectronic oscillator, interrogation

## Abstract

The design of a refractometric sensing system for liquids analysis with a sensor and the scheme for its intensity interrogation combined on a single photonic integrated circuit (PIC) is proposed. A racetrack microring resonator with a channel for the analyzed liquid formed on the top is used as a sensor, and another microring resonator with a lower Q-factor is utilized to detect the change in the resonant wavelength of the sensor. As a measurement result, the optical power at its drop port is detected in comparison with the sum of the powers at the through and drop ports. Simulations showed the possibility of registering a change in the analyte refractive index with a sensitivity of 110 nm per refractive index unit. The proposed scheme was analyzed with a broadband source, as well as a source based on an optoelectronic oscillator using an optical phase modulator. The second case showed the fundamental possibility of implementing an intensity interrogator on a PIC using an external typical single-mode laser as a source. Meanwhile, additional simulations demonstrated an increased system sensitivity compared to the conventional interrogation scheme with a broadband or tunable light source. The proposed approach provides the opportunity to increase the integration level of a sensing device, significantly reducing its cost, power consumption, and dimensions.

## 1. Introduction

Nowadays, sensing systems for label-free determination of the substance concentration in a liquid play a significant role in modern medicine [1], experimental biology [2], and applied chemistry [3]. In the general case, the structure of a photonic sensing system includes an optical sensor circuit, a light source, a detector, and an interrogation and control circuit. The technological maturity of integrated photonics makes it possible to effectively implement the individual elements of such a system in an integrated form to create high-performance lab-on-a-chip devices. At the same time, increasing the integration level with the required stability of the characteristics is an obvious way to implement refractometric sensing systems, which are in demand in a wide range of applications.

Among the various approaches to implementing refractometry systems, phase-sensitive circuits are the most attractive in terms of technological viability and efficiency. Widely used schemes for detecting changes in the physical properties of an analyte include fiber (FBG) and waveguide (WBG) Bragg gratings [4,5,6,7], waveguide Mach–Zehnder interferometers (MZI) [8,9,10], and microring (MRR) or microdisk (MDR) resonators [11,12,13].

FBG sensors have proven to be very simple to implement yet highly efficient structures for environmental measurements [14,15]. A shift in the resonant wavelength can be caused by a grating deformation or various types of interactions between the analyte and the grating structure. These interactions are achieved by forming a phase shift in the lattice [5] or integrating a Fabry–Perot interferometer, which allows for obtaining a sensor sensitivity of up to 1210.49 nm per refractive index unit (RIU) [7].

In addition to fiber sensors, high performance can be obtained using integrated sensors due to their ability to fabricate MZI, MRR, and WBG on photonic integrated circuits (PICs) using technological platforms such as silicon nitride (Si_3_N_4_) or silicon-on-insulator (SOI). SOI structures are most often used for operation in the near-infrared and mid-infrared ranges due to their relatively low propagation losses and small allowed bending radii of the Si waveguides. These structures also have a large set of components that are supported for monolithic integration, including fast diode phase modulators and photodetectors [16].

Among all existing types of sensors on PICs, MRR is the most used [17]. MRRs have great potential to implement label-free detectors because they are fast, energy-efficient, and have a significant sensitivity to detect biomolecules in liquids and gases [12,13]. MRR fabrication technology is widely applied, and silicon MRRs can be integrated into a microfluidic system [11]. The principle of operation is based on a shift in the resonant wavelengths of the ring caused by a change in the refractive index (RI) near the sensor, which occurs when the studied parameter of the substance varies.

The sensitivity of MRR sensors currently varies from 70 to 490 nm/RIU depending on the type of waveguide and the selected operating mode [11]. Specifically, for the MRR sensors on SOI, sensitivity does not exceed 100 nm/RIU [18,19]. MRR cascading [13,20], subwavelength grating microring [21,22], or phase-shifted Bragg grating microring [23] have been proposed to increase the sensitivity.

The main problem in utilizing MRRs as sensors is a need for a tunable mode-hop-free laser or a broadband source to excite the resonances within the structure. Hybrid integration of a semiconductor laser directly onto a chip is a technologically feasible but rather expensive solution. In this context, the generation of broadband radiation directly on a PIC could be a solution capable of circumventing these technological limitations.

The output characteristics of a sensing system are greatly determined by the efficiency of the interrogation scheme. The typical solution for assessing a change in the optical spectrum is using commercial interrogators based on tunable lasers [24]. However, the pricey equipment and the system bulkiness are among the main disadvantages of this approach. For the practical application of optical sensors, indirect methods have been developed to analyze the optical emission spectrum for high-speed interrogation with high resolution [25]. Depending on the analyzed parameters of the output electrical signal, three methods of interrogation are distinguished: by intensity, when the photocurrent power is measured [26]; by frequency, when the frequency of the signal at the output of the photodetector is estimated [27,28]; and by time, when the shape of the output signal envelope is analyzed [29].

Intensity interrogation can provide fast sensor scanning, but the radiation source dispersion or the low stability of the optical source affects the measurement accuracy [29]. Frequency interrogation can be implemented by an optoelectronic oscillator (OEO), where the generated frequency depends on the resonant characteristics of the sensor. High-precision measurements can be achieved by the ultra-high wavelength resolution of the OEO (up to 360 fm [28]). However, this method requires additional devices for measuring the microwave signal frequency. Time interrogation enables a high scanning rate (up to 48.6 MHz [29]), but the resolution of such systems is inferior to those described above. In addition, the accuracy of the signal shape estimation is related to the capacity of the analog-to-digital converter (ADC). Thus, for integrated implementation, it is preferable to use intensity interrogation because its limitations are determined mainly by the shortcomings of the radiation source.

PIC-based interrogation systems are extremely interesting in sensing applications [30], especially in the case of wearable devices and medical diagnostics [24]. In addition, monolithic integration of the interrogation system and the sensor into a single miniature chip will potentially reduce losses and the footprint, consequently creating ultra-compact sensing systems with high performance. However, to our knowledge, PIC-based sensing systems with such an integration level have not been previously discussed.

In this paper, we propose for the first time, to our knowledge, the design of a sensing system for liquids refractometry that is suitable for the monolithic integration of both the sensor and interrogator parts (except only the light source) on a single SOI PIC. Two different design options are considered, including an external broadband light source and an OEO. The latter allows for an off-the-shelf, single-mode laser in the intensity interrogation scheme.

The remainder of this paper is organized as follows: in the second section, the proposed schemes of an integrated sensing system and the main design problems are discussed; the third section describes the MRR-based sensor and interrogator design features and characteristics; the fourth section presents the simulation results for the intensity interrogation scheme based on an MRR and a broadband radiation source, as well as for the scheme based on an MRR and an OEO source; in the fifth section, the application aspects of the designed interrogation system together with an FBG and a phase-shifting FBG (PS-FBG) are considered; and in the Discussion section, the possible options for PIC-based implementation of the proposed system on different technological platforms are considered, as well as the modeling results for the influence of the waveguide width deviation on the system operability.

## 2. PIC-Based Sensing System Concept and Operation Principle

### 2.1. Microring-Based Sensing System with a Broadband Optical Source

The structure of the proposed sensing system is shown in Figure 1. The input port of the microring racetrack resonator, which acts as a sensor, receives a broadband signal from an optical source. The effective RI of the MRR’s waveguide depends on the RI of the medium surrounding the MRR, which determines the position of the resonant peaks in the transmission spectrum at the MRR’s drop port. The proposed system is designed to detect and measure changes in the properties of aqueous solutions. To do this, a microfluidic channel can be connected to the MRR sensor by analogy with [11].

To trace the sensor resonant wavelength without using a mode-hop-free tunable laser or an optical spectrum analyzer, which are non-compliant with planar integration, we propose an interrogation scheme by intensity, where another MRR is utilized for the sensor interrogation.

In this case, the interrogation process is a transformation of a change in the resonant wavelength at the input port of the interrogator into the intensity of optical power at the through and drop ports of the interrogator (Figure 2).

The signal at the output of the sensor’s drop port is determined by its resonant characteristic. The MRR resonant peak width at the half power level (FWHM) is given by:(1)$$FWHM=\frac{\left(1-r_{1}r_{2}a\right)\lambda_{res}^{2}}{\pi n_{g}L\sqrt{r_{1}r_{2}a}}$$
where *a* = exp(−α*L*/2) is the ring amplitude transmission coefficient; α is the ring waveguide power attenuation coefficient; *r*_1_ and *r*_2_ are the amplitude transmission coefficients of the directional couplers to the in-through and add-drop waveguides, respectively; λ*_res_* is the MRR resonant wavelength (λ*_res_* = *Ln_eff_*/*m*, where positive integer *m* is the resonance number); *L* is the MRR circumference; and *n*_g_ and *n_eff_* are the values of the waveguide group and effective refractive indices, respectively. The optical power transmission coefficients at the through and drop ports of the MRR are determined, respectively, by the equations:(2a)$$T_{p}=\frac{I_{pass}}{I_{input}}=\frac{r_{2}^{2}a^{2}-2r_{1}r_{2}a\cos\varphi +r_{1}^{2}}{1-2r_{1}r_{2}a\cos\varphi +{\left(r_{1}r_{2}a\right)}^{2}}$$
(2b)$$T_{d}=\frac{I_{drop}}{I_{input}}=\frac{\left(1-r_{1}^{2}\right)\left(1-r_{2}^{2}\right)a}{1-2r_{1}r_{2}a\cos\varphi +{\left(r_{1}r_{2}a\right)}^{2}}$$

As mentioned in the Introduction, one of the main drawbacks of the intensity interrogation system is its dependence on the instability of the optical source output power. To address this shortcoming, we propose using the relative value defined as: (3)$$P=10\mathrm{lg}\left(\frac{p_{drop}}{p_{drop}+p_{thru}}\right)$$
where *p_drop_* and *p_thru_* are the output powers at the sensor MRR’s drop and through ports, respectively. Thus, the proposed interrogation scheme estimates the power level, *p_drop_*, at the drop port of the interrogator MRR relative to its total output power (*p_drop_* + *p_thru_*). These powers may be calculated using the MRR transmission spectrum (2a, 2b). In this case, considering that *p_thru_* = *P_in_T_p_* and *P_in_* = *P_in_*_0_ + Δ*P_in_*, the relative power will be determined as follows:(4)$$P=10\mathrm{lg}\left(\frac{T_{d}}{T_{d}+T_{p}}\right)$$

Thus, the measured relative power does not depend on fluctuations in the optical source output power.

### 2.2. Matching the Frequency Characteristics of the Sensor, Interrogator, and Broadband Source

An analysis of Figure 2 allows us to formulate the essential requirements for the sensing system consisting of the two MRRs. First, the FWHM of the sensor MRR should be substantially smaller than the FWHM of the interrogator MRR to provide a greater sensitivity and measurement range of the interrogator. Practically, the FWHM of the sensor MRR can be reduced by increasing its circumference without changing the coupling coefficient in directional couplers, e.g., by inserting straight waveguide sections between the MRR half rings (using racetrack-MRR). Therefore, the quality factor, *Q*, of the interrogator MRR, defined as:(5)$$Q=\frac{\lambda_{\mathrm{res}}}{\mathrm{FWHM}}$$
should be lower than that of the sensor. The actual relationship between the FWHMs of the sensor and interrogator MRRs should be determined for specific application scenarios and the possible range of the sensor’s resonant wavelength change. Ideally, this range should be completely located within the monotonic section of the utilized resonant peak in the interrogator transmission spectrum.

Second, it should be considered that the MRR resonances occur periodically with a free spectral range (FSR) determined by:(6)$$\mathrm{FSR}=\frac{\lambda^{2}}{n_{g}L}$$ and thus, the resonant peaks in the sensor and the interrogator transmission spectra must intersect only in the wavelength range corresponding to the operating range of the sensing system (see Figure 2). In practice, this condition can be fulfilled in two ways. The first option is to select the bandwidth of the optical source to be equal to the half FWHM of the interrogator’s resonance and its central wavelength coinciding with the middle of the monotonic section of that resonance. The second option is to choose the geometry of the sensor and interrogator MRRs so that the resonant peaks, except for the analyzed sensor peak and the corresponding operating interrogator peak, would not overlap in a sufficiently wide wavelength range exceeding the bandwidth of the optical source. The second option is universal and practically achievable in terms of selecting a source, so let us consider it in more detail. On the one hand, the large radius of the sensor ring ensures its high-quality factor and, consequently, its sensitivity. On the other hand, increasing the radius of the sensor ring leads to a decrease in its FSR, which can lead to the response of the interrogator to be closely spaced sensor resonant peaks. An example of such a negative overlap is shown in Figure 3.

The left peak of the interrogator transmission spectrum is situated on the decline region of the sensor’s resonant characteristic, whereas the right peak is on the rise region. Then, with an increase in the sensor resonant wavelength in response to the analyte RI change, the signal from the left peak at the interrogator’s drop port will increase, and the signal from the right peak will decrease, which will prevent correct measurements. To reduce this probability, the FSR of the interrogator must be suitably large, which also provides the required lower quality factor, thus increasing the RI measuring range. It should be considered that commercially available optical sources have either a very narrow band (CW lasers) or, in the case of SLD sources, provide a bandwidth of 30–40 nm at a 3 dB level [31]. A narrower band (up to 20 nm) can be obtained by installing an additional filter based on AWG (arrayed waveguide grating) [32] or FBG at the source output. Such a filter can be either external, combined with the source, or integrated directly into the sensing system PIC (in the case of using AWG). A larger source bandwidth increases the probability that several interrogator’s resonant peaks fall into the source band and the likelihood of a destructive overlap between the sensor and the interrogator characteristics. Thus, the choice of MRR dimensions is determined not only by a trade-off between the system sensitivity and the range of measured RI values but also by the characteristics of the used optical source.

### 2.3. Microring-Based Sensing System with an Optoelectronic Oscillator

The general problem with intensity interrogation is the need for a broadband optical source [26] or a cascade of narrow-band sources [6], which complicates the integrated implementation of the interrogator and increases its cost. Thus, it is crucial to consider how to obtain a feeding optical signal on a PIC and how to reduce its influence on the sensing system characteristics. Let us consider a Mach–Zehnder phase modulator covered by a positive feedback loop as a source. Such a system performs the function of an optoelectronic oscillator (OEO) [33]. Figure 4 shows the structure of the proposed PIC-based interrogator, using a frequency self-adjustable OEO as an optical radiation source. It consists of two key components: a phase MZM and MRR, where the latter performs as a notch filter at the drop port and a bandpass filter at the through port. It should be noted that the photodetector can be made either as a discrete external device or implemented on a PIC, as shown in Figure 4. The microwave elements are shown tentatively. They may be mounted on the printed circuit board (PCB) as discrete planar devices or implemented as a monolithic microwave integrated circuit. The electrical circuit design is out of the scope of this paper.

The operating principle of an OEO was described in [33]. To start the generation, the narrow-band continuous-wave (CW) optical signal from an external laser diode is used as the carrier signal for the phase Mach–Zehnder modulator (MZM). At the output of the phase MZM, a phase-modulated (PM) optical signal is generated and sent to the MRR sensor, which serves as an optical notch filter. A feature of the PM signal spectrum is the phase opposition of the spectral components, which are symmetrically relative to the carrier. Therefore, when a PM signal is applied to the photodiode (PD), the antiphase components compensate each other, and the electrical signal at the output of the PD is determined only by the intensity of the optical carrier. However, when the notch filter suppresses one of the spectral components of the PM optical signal at the PD input, one of the spectral components becomes uncompensated. This causes a microwave signal to appear at the PD output. The frequency of this signal is determined by the difference between the CW wavelength and the resonant wavelength of the notch filter (i.e., the sensor MRR):(7)$$f=\left|\frac{c}{\lambda_{CW}}-\frac{c}{\lambda_{MRR}}\right|$$
where *c* is the velocity of light in vacuum; and λ*_CW_* and λ*_MRR_* are the laser source wavelength and the resonant wavelength of the sensor, respectively. The microwave signal through the amplifier is fed to the input of the phase MZM, thereby closing the positive feedback loop.

Following the described operation principle, the OEO source bandwidth at the sensor drop port will be very narrow, as it is defined by the width of one spectral component in the signal, phase-modulated by a narrow-band microwave signal (the confirming simulation results are given in Section 4.2). In this case, the main wavelength of this optical signal is unambiguously related to the sensor transmission spectrum and changes along with it. Thus, the scheme shown in Figure 4 allows the generation of the input signal for the intensity interrogator without using a broadband optical source.

## 3. Sensing System Elements

### 3.1. Microring-Based Sensor

The layout of a label-free MRR sensor for liquid analysis is shown in Figure 1. It represents a silicon racetrack MRR on a SiO_2_ buried-oxide layer, operating in a real-time regime. Hereinafter, the material dispersion in the waveguides is modeled by using the RI wavelength dependencies of the materials from [34]. The following MRR sensor geometric parameters were set: an outer radius set at 18 μm, the straight waveguide length of the racetrack set at 3 μm, the gap in the directional coupler of the in-through ports set at 0.18 μm, the gap in the directional coupler of the add-drop ports set at 0.25 μm, and the Si waveguide height/width set at 0.22/0.4 μm. The frequency response of the MRR sensor was calculated using the FDTD (Finite Difference Time Domain) method in the Ansys Lumerical software as follows: a simulation time of 7000 fs, a mesh accuracy of 3, and a minimum mesh step of 0.25 nm. The calculation was carried out for the TE-mode. The dispersion, loss, effective RI, and group RI were defined using the Finite Difference Eigenmode Solver the in Lumerical MODE. The MRR calculation method was employed as presented in [35,36].

The gap value for the in-through waveguide was selected based on the critical coupling criterion [37]. Using FDTD, the attenuation in the MRR waveguide and the coupling coefficients of the directional couplers were calculated. The resulting gap value for the in-through waveguide equals 0.18 μm. The presence of an add-drop waveguide introduces additional losses into the MRR system caused by the transfer of energy from the ring to the drop port. Therefore, the add-drop port coupler’s gap value was chosen to have a high Q-factor in the under-coupling regime, providing enhanced sensitivity [17]. The resonant properties of the MRR were evaluated using FWHM (Equation (1)) and Q-factor (Equation (5)) using the following values: the ring transmission was measured using FDTD to be *a* = 0.971; the amplitude coupling coefficients to the in-through and add-drop waveguides are *r*_1_ = 0.935 and *r*_2_ = 0.961, respectively; the MRR circumference is *L =* 2π*R* + 2*L_s_* (*L_s_ =* 3 μm is the length of the straight waveguide in the racetrack, *R* = 18 μm is the ring radius); and *n*_g_ = 4.55 and *n_eff_* = 2.21 are the values of the waveguide group and effective RIs, respectively, determined numerically using the Ansys Lumerical MODE software R1.4 2021.

Figure 5 shows a fragment of the transmission spectrum at the through and drop ports of the sensor for the minimum value of the analyte RI (1.311). According to Figure 5, with the gap values set as described above, the FWHM equals 190 pm (whereas in the case of equal gap values for both couplers of 0.18 μm, the FWHM would be 270 pm). Thus, an increase in the gap with the add-drop waveguide to 0.25 μm reduced the coupling coefficient and FWHM, providing the calculated quality factor Q ≈ 8000. However, a further gap expansion leads to a decrease in the power at the drop port, making it more difficult to register.

As mentioned in the Introduction, a variation in any parameter of a liquid flowing over the sensor surface (e.g., composition, concentration, or temperature) shifts the resonant wavelengths of the MRR. Figure 6 demonstrates this process depending on the analyte RI sweeping from 1.311 to 1.315 with ∆*n* = 0.001 (the analyzed resonant peak around 1537.4 nm is shown). To obtain the sensor MRR transmission spectrum depending on the analyte RI, the FDTD calculation of the sensor MRR was redone for each analyte RI value above the MRR waveguide within the channel area (see Figure 1). The resonant wavelength shift, ∆λ, shows almost linear behavior, with ∆λ = 0.11 nm at ∆*n* = 0.001. From these results, the sensor sensitivity, *S*, can be evaluated as follows:(8)$$S=\frac{\Delta \lambda }{\Delta n_{eff}}=110\;\frac{\mathrm{nm}}{\mathrm{RIU}}$$

The resulting sensitivity value slightly exceeds the results presented in [18], where a design with different gaps in the straight and add-drop couplers was also used. This improvement can be explained by the method applied for calculating the resonator’s design parameters. At the same time, because we used a single MRR sensor in the simulation, the sensitivity is lower than in [21,22,23].

### 3.2. Microring-Based Interrogator

To fulfill the requirements of the interrogator transmission spectrum discussed in Section 2, the parameters of the interrogator’s MRR were defined as follows: the outer radius was set at 10 μm, the gap in both directional couplers was set at 0.1 μm, and the Si waveguide height/width were set at 0.22/0.4 μm. The interrogator MRR frequency response and corresponding Q-factor and FWHM values were calculated using the FDTD method in the Ansys Lumerical software, similar to the sensor MRR. The corresponding Q-factor of the interrogator’s MRR read ~1400, and the FWHM equaled 1.1 nm.

Figure 7 shows a fragment of the interrogator MRR transmission spectra at the through and drop ports.

## 4. Sensing System Simulation

### 4.1. System Simulation with a Broadband Source

A simulation of the whole sensing system was carried out following the scheme shown in Figure 1 in the Lumerical INTERCONNECT environment. The transmission spectra of the sensor and interrogator (for the minimum considered analyte RI of 1.311) are shown in Figure 8a. The resonant peaks of the sensor and interrogator MRRs overlap at the wavelength of 1537.37 nm (this resonant peak of the sensor is selected as the analyzed one); the next resonant peak of the sensor is at 1541.91 nm (so, the FSR equals 4.54 nm), and the previous resonant peak is at 1532.83 nm. There is a partial overlapping of the resonant peaks of the sensor and interrogator at 1546.52 nm, and the next resonant peak of the interrogator is at 1553.4 nm (the FSR of the interrogator equals 8.07 nm). Therefore, the broadband source may have a spectrum width of up to 20 nm, but the wavelengths from 1533 nm to 1535 nm and from 1550 nm to 1553 nm are useless because the transmission of the interrogator is very close to zero within those ranges. Therefore, to accelerate the simulation, the broadband source was emulated with an optical network analyzer (ONA) unit with a wavelength range from 1535 nm to 1550 nm, and the output powers at the interrogator PDs were calculated according to the equation:(9)$$P_{pd}=Z_{out}{\left(\overset{\lambda_{max}}{\underset{\lambda_{min}}{\int }}P_{l}\left(\lambda \right)R{\left(\lambda \right)}^{2}d\lambda \right)}^{2}$$
where *P_l_*(λ) is the optical power at the PD input; *R*(*λ*) is the PD responsivity; *Z_out_* is the PD load resistance; and *λ*_min_ and *λ*_max_ are the minimum and the maximum wavelengths of the broadband source, respectively (1535 nm and 1550 nm, respectively, in our case).

The source signal enters the in-port of the sensor, and when the resonance condition is met, the signal appears at the sensor drop port and enters the in-port of the interrogator. Figure 8b shows how the optical power intensity at the drop port of the interrogator changes depending on the analyte RI. Suppose the analyzed resonant wavelength of the sensor and the utilized resonant wavelength of the interrogator coincide. In that case, there will be a maximum transmission coefficient and maximum intensity of the optical signal at the drop port of the interrogator. If the analyzed resonance of the sensor hits within the FSR of the interrogator, then the transmission coefficient will be minimal. Almost all the optical power will pass to the through port of the interrogator.

To plot the dependence of the relative power at the system output on the analyte RI, we first analyzed the optical intensity at the through and drop ports of the sensor. The results for each port were separately converted into units of electrical power according to Equation (9), considering the wavelength-dependent responsivity of the PD (Figure 9a). The typical SOI-based structure described in [38] was used for the PD numerical model. Then, for each value of the RI, the relative power at the interrogator drop port was calculated using Equation (3).

The results are shown in Figure 9b. As can be seen from the graph, a change in the RI of 0.001 leads to a change in the relative power of at least 1.35 dB, which makes it possible to estimate the minimum aggregate sensitivity of the entire system by analogy with Equation (8):(10)$$S_{\min}=\frac{\Delta P_{\min}}{\Delta n}=1350_{}\frac{\mathrm{dB}}{\mathrm{RIU}}$$
The averaged system sensitivity over the entire measurement range equals $\bar{S}=1570\;\mathrm{dB}/\mathrm{RIU}$.

The simulation results confirm the validity of the requirements outlined in Section 2. The choice of the MRR geometry provided a higher Q-factor for the sensor and a lower Q-factor for the interrogator. The ratio of the sensor and interrogator FSRs made it possible to use a broadband source (with spectrum width up to 20 nm) without the negative resonances overlapping effect. Of course, it is possible to expand the bandwidth of a broadband source with a decrease in the interrogator MRR radius, but this would lead to a further decrease in its quality factor and the slope of the resonant characteristic, which would reduce the system sensitivity.

### 4.2. Simulating the System with Optoelectronic Oscillator as a Source

Let us consider the sensing system scheme with the OEO utilized as a radiation source, according to Figure 4. The possibility of an integrated OEO design on the SOI platform was demonstrated in [39]. The simulation was performed in the Lumerical INTERCONNECT environment with the following circuit parameters: a CW laser carrier wavelength of 1538 nm; a CW laser output power of 2 mW; a microwave amplifier gain of 65 dB; the PD model is the same as in Section 4.1; the PD dark current of 60 nA; the PD, amplifier, and MZM RF bandwidth *f*_max_ = 50 GHz; and the MZM *V_π_* = 4 V. At the current stage of research, the idealized phase MZM transfer function is used without parasitic amplitude modulation because the real phase modulator characteristics will be strongly dependent on the parameters of the photonic integration platform used for the system implementation. The parameters of the MRR sensor and MRR interrogator are given above in Section 3. The choice of the CW laser wavelength is determined by the range of the analyzed sensor resonance wavelengths and the allowable microwave signal frequency, which is determined by the frequency characteristics of the PD, modulator, and amplifier. To set the value of the laser carrier wavelength for analyzing a certain sensor resonance peak, it is necessary to define the maximum value of the microwave signal frequency, *f*_max_. Then, it is easy to convert Equation (7) into an expression for determining the required optical carrier wavelength. For definiteness, we assume that the wavelength of the CW laser is greater than the analyzed resonance wavelength. In this case, the highest value of the microwave signal frequency will correspond to the lowest value of the MRR resonant wavelength, λ*_MRR_*_min_. Then:(11)$$f_{\max}=\frac{c}{\lambda_{MRR\min}}-\frac{c}{\lambda_{CW}}$$
(12)$$\lambda_{CW}=\frac{c\cdot \lambda_{MRRmin}}{c-f_{\max}\lambda_{MRRmin}}$$

Figure 10 shows the simulated optical spectra at the through and drop ports of the sensor MRR. The dependence of the relative power at the interrogator drop port on the analyte RI, obtained from the simulation, is shown in Figure 11.

It can be seen that a change in the RI of 0.001 leads to a change in the output relative power of at least 2.51 dB, which makes it possible to evaluate the minimum aggregate sensitivity of the entire sensing system by analogy with Section 3.1:(13)$$S_{\min}=\frac{\Delta P_{\min}}{\Delta n}=2510_{}\frac{\mathrm{dB}}{\mathrm{RIU}}$$
The averaged system sensitivity, in this case, is $\bar{S}=3935\;\mathrm{dB}/\mathrm{RIU}$.

Thus, when using an OEO, the system’s sensitivity is much higher than when using an optical broadband source. However, the use of an OEO in the described intensity interrogation system has some specifics. First, the interrogation wavelength range is limited by the RF bandwidth of the phase modulator, RF amplifier, and PD (see Equation (7)). For example, at the maximum OEO generation frequency of 30 GHz, the interrogation range will be 235 pm, and when the frequency range is extended to 50 GHz, the interrogation wavelength range will increase to 395 pm. At the same time, this limitation in the analyzed spectrum width removes the problem mentioned above regarding the unintended mixing of the useful signal from the analyzed resonant peak of the sensor MRR and the parasitic signals from its other resonant peaks.

Second, the use of an OEO causes additional delays. Figure 12 shows that it takes about 5 ns to start generating in a steady-state operational mode. During this time, the output of the interrogator will not be reliable.

Moreover, the scheme with an OEO is anticipated to be more power-consuming due to the need for optoelectronic and electro-optic conversions and microwave signal amplification in the feedback loop.

Despite the shortcomings described, the interrogation scheme using an OEO also has obvious advantages. First, there will be only one resonant peak in the signal spectrum at the drop port of the MRR sensor (Figure 10b); second, the linewidth of the optical spectrum generated by the OEO is significantly narrower than the resonance FWHM of the MRR used as a sensor. Thus, the optical spectrum width, according to the simulation, was less than 10 pm, even though the FWHM of the sensor is 190 pm. These advantages avoid the negative overlap of adjacent peaks of the sensor transmission spectrum described in Section 2 and, as can be seen from the simulation results, increase the sensitivity of the sensor system.

## 5. Bragg-Grating Interrogation Using an MRR-Based Interrogator

Fully integrated sensing systems are of interest, as described in the Introduction. However, in monitoring extended objects (i.e., pipelines and bridges), it is advisable to create quasi-distributed sensor systems in which the sensors and the interrogators are geographically spaced. Today, in such systems, FBG sensors are widely used, so it makes sense to also evaluate the compatibility of the proposed PIC-based interrogator with FBG. The simulation was carried out for a typical grating using the parameters presented in Table 1. The transmission and reflection spectra of the FBG are shown in Figure 13. For representativeness, an ONA was used as an input optical signal source. The interrogator output signals were calculated by integrating the ONA output signals over the spectrum width of the broadband source. The spectral band of the broadband source was assumed to be the same as in the model with the MRR sensor (1535–1550 nm).

The modeling has shown that the scheme using an OEO as a source cannot work properly with standard FBG due to the relatively wide FBG’s output spectrum. However, an OEO can be applied with appropriately designed fiber grating sensors, e.g., a narrow-band phase-shifting Bragg grating (PS-FBG).

The MRR-based interrogation scheme was modeled with FBG replacing the MRR sensor. Figure 14a shows the first operation option when the FBG transmission signal is processed, whereas Figure 13b demonstrates the second option when FBG is connected through an optical circulator to operate with the reflected signal.

Figure 15a,b show the interrogator drop port transmission spectra and demonstrate how the transmission changes depending on the shift of the Bragg wavelength. The simulated relative power at the drop port of the interrogator versus the resonant wavelength of the FBG is provided in Figure 15.

The sensitivity S_BG_ can be calculated as follows:(14)$$S_{BG}=\frac{\Delta P}{\Delta \lambda }$$
where Δ*P* is the difference between the neighboring values of the relative power levels at the interrogator drop port and Δλ is the corresponding change in the resonant wavelength. According to Equation (14), the minimum system sensitivity for analyzing the transmitted and reflected light are 7.2 dB/nm and 2.27 dB/nm, respectively. The average sensitivity may be calculated as the ratio between the relative power change and the resonant wavelength variation; the average sensitivity equals 13.7 dB/nm and 2.8 dB/nm for analyzing the transmitted and reflected light, respectively.

By comparing these two interrogation options for an FBG sensor (Figure 16), it can be noticed that the sensitivity turns out to be higher when operating with the reflected light. At the same time, for the option that analyzes the transmitted light, the sensitivity is flatter over a wider range of the FBG’s resonant wavelength shifts. This result can be explained by the shape of the interrogator’s transmission spectrum. When operating in reflected light, the FBG works as a bandpass filter, passing the signal to the interrogator near the high-steepness region of its transmission spectrum. When operating in the transmitted light, the FBG works as a notch filter, and the wavelengths at the in-port of the interrogator fall into the low slope region of the filter characteristic.

Due to the relatively wide spectrum width of the reflected signal, it is not possible to use a classical Bragg grating in an interrogation scheme with an OEO. However, the PS-FBG has a very narrow stopband spectrum when analyzing the reflected light. We simulated the interrogation system with the PS-FBG (parameters are given in Table 2) following the scheme shown in Figure 17a. The grating spectrum is shown in Figure 17b. The simulated relative power at the drop port of the interrogator versus the resonant wavelength of the PS-FBG is provided in Figure 18. The average sensitivity of the sensing system in the model reads about 37 dB/nm.

In turn, the PS-FBG cannot be used in the intensity interrogation scheme when operating in transmitted light due to the wide output of the optical spectrum. However, its use when analyzing reflected light (by analogy with Figure 14b) is possible.

The simulated relative power at the drop port of the interrogator versus the resonant wavelength of the PS-FBG in the scheme broadband source is provided in Figure 19. The corresponding average sensitivity of the sensing system is about 55 dB/nm.

Although the scheme with the broadband source provides higher system sensitivity, it should be underscored that it requires a more complicated source with a certain bandwidth, which may be hard to realize on a PIC. System sensitivity for the interrogation scheme with an OEO is lower but remains sufficient to distinguish the output electrical signal variations using modern electronic components.

## 6. Discussion

The proposed integrated sensing system can be fabricated on the SOI platform. Other commonly used technological platforms such as silicon nitride, indium phosphide (InP), and polymer-based have several limitations [40], making them less suitable for the fabrication of the considered scheme.

Thus, on the Si_3_N_4_ platform, modulators can be implemented with heaters only, but the thermo-optical effect provides operation times down to 30 µs [41], which limits the performance of the interrogation scheme with an OEO. Meanwhile, for polymer-based platforms, modulation speeds of about 250 ns can be achieved with a stress-optical effect [42], and in commercially available InP platforms, the phase modulators based on p-i-n hetero-junction can provide a modulation bandwidth >20 GHz [43]. However, all aforementioned platforms are of lower contrast [44] compared to SOI, which leads to an increased radius of the MRRs. Owing to this, the FSR values become too small, which significantly complicates the system design required to avoid unintended interference from the side resonant peaks.

For a fully integrated system, laser source implementation on a chip is required. Only the InP platform can provide the full range of photonic devices. However, in the standard substrate-based InP platform, there are no small bend radii of the waveguides. The waveguide radii similar to those of the SOI platform can be achieved on the novel InP membrane platform [45], which is currently under development and appears promising for full integration of the considered sensing system. Alternatively, the hybrid integration of InP elements could be used to overcome this problem. For instance, passive elements can be implemented using an SOI or Si_3_N_4_ process and then connected to the InP chip with active elements via 45° mirrors or butt-joint coupling [40]. In addition, flip-chip integration is available for the silicon nitride platform [46]. These integration approaches provide insertion losses from 1 dB to 2 dB [40,46] that are lower than when using discrete elements connected via optical fiber.

A promising solution is using monolithic hybrid platforms [47,48], which provide a smaller footprint and better reliability than conventional hybrid integration methods. It is worth noting that both platforms have optical sources with a bandwidth higher than 20 nm [48,49].

One of the problems in implementing PIC-based devices is the inaccuracies in the feature size of the fabricated chips. For the scheme under consideration, the deviation in the width of the waveguides appears to be the primary type of fabrication inaccuracy that can affect the system performance, as it changes the effective and group indices of the waveguides and the coupling coefficients of the directional couplers in MRRs. According to the experimental data presented in [35], the intra-wafer standard deviation of the waveguide width for the typical SOI fabrication process based on 248 nm photolithography is about 3.9 nm. In the commercially available state-of-the-art 193 nm immersion lithography and dry etch fabrication process, the intra-wafer waveguide width standard deviation can be as low as 2.5 nm [50]. However, according to [51], for different wafers, the standard deviation of waveguide width can reach 6.4 nm. Based on the wafer maps of the waveguide geometry variations, their influence on the performance of certain integrated components can be analyzed, and the corresponding fabrication yield can be predicted and optimized, as presented in [52].

We have analyzed the possibility of maintaining the system operability with errors in the waveguide width up to 8 nm, which is guaranteed to consider the characteristics of available platforms. The performed analysis showed (Figure 20) that the sensitivity of 110 nm/RIU is retained; although, changing the waveguide width significantly affects the resonant frequencies of the MRR sensor and the transfer function at the drop port. The change in the position of the resonant frequencies of the sensor MRR relative to the frequency characteristics of the interrogator can be compensated using a built-in heater for the interrogator MRR (Figure 1 and Figure 4). The same method can be applied in the case of FBG sensors interrogation to compensate for their possible Bragg wavelength deviation, which practically does not exceed tenths of a nanometer. The systematic amplitude and phase inaccuracies during the FBGs inscription and their mitigation have been described in the literature [53,54]. In a lower-quality interrogator MRR, the waveguide width deviation only leads to a shift in the resonant frequencies (Figure 21), which can also be corrected using a heater. Changes in the FWHM and Q-factor are negligible. Therefore, the sensitivity of the entire system will not change. Thus, the proposed schemes will remain operable within a practically relevant range of fabrication errors in the waveguide width.

## 7. Conclusions

The concept of a sensing system based on microring add-drop resonators, suitable for integration on the SOI platform, is proposed. It includes a sensor based on the racetrack MRR, which, according to simulation results, has a sensitivity of 110 nm/RIU and the interrogator based on the MRR with a lower Q-factor.

A novel scheme for generating an input optical signal for feeding the sensor using the OEO is also proposed. The key feature of this scheme is that it does not require either an external broadband source or a tunable laser for its operation. At the same time, the average sensitivity of the system with an OEO is 3935 dB/RIU, which is even higher than that for the configuration with a broadband or tunable source (1570 dB/RIU). Despite the limitation of the sensor’s transmission spectrum width that can be analyzed, this scheme allows for interrogating the sensor targeted exclusively on the analyzed resonant peak and avoiding crosstalk from the other resonant peaks. Noteworthy, the transient process of the OEO causes an additional delay in measurements. Yet, the simulation showed that the introduced delay is lower than 10 ns, which is relevant only at sampling rates exceeding 100 MHz.

As modeling revealed, the interrogation scheme with an OEO cannot be applied with typical FBG-based sensors due to their relatively wide stop band. Therefore, system performance with the FBG sensor was evaluated using a broadband source. The scheme provides classical FBG interrogation in reflected and transmitted light. The system sensitivity in the former case is considerably higher; although, it is less monotonic because the FBG output spectrum is wide and may cover the interrogator transmission spectrum regions with different slopes. At the same time, interrogation in transmitted light does not require an optical circulator, providing a less expensive and more compact scheme suitable for full integration. That fact makes it a good candidate for applications with undemanding high sensitivity. Nevertheless, the OEO could be a promising integrable solution as a source operating with narrow-band gratings, such as PS-FBG. For such gratings, simulation was performed with a broadband source and an OEO. The results have shown that the former provides a higher system sensitivity but requires a more complicated source that may be difficult for PIC-based realization; the latter provides a sufficient system sensitivity when using it with a control circuit realized on modern electronic components.

## Figures and Tables

**Figure 1 sensors-22-09553-f001:**
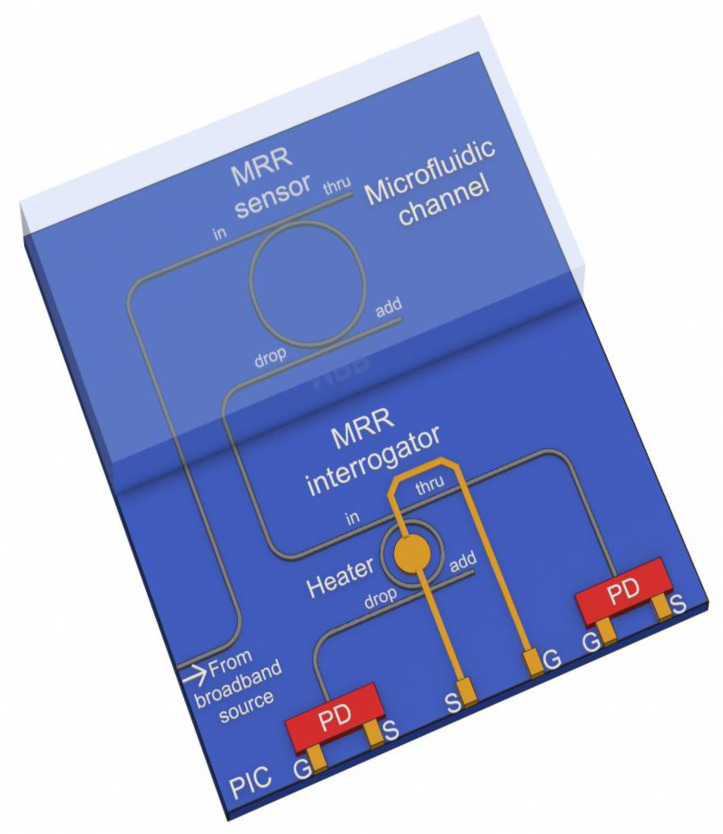
3D design draft of the proposed integrated MRR-based sensing system with a broadband source (not to scale).

**Figure 2 sensors-22-09553-f002:**
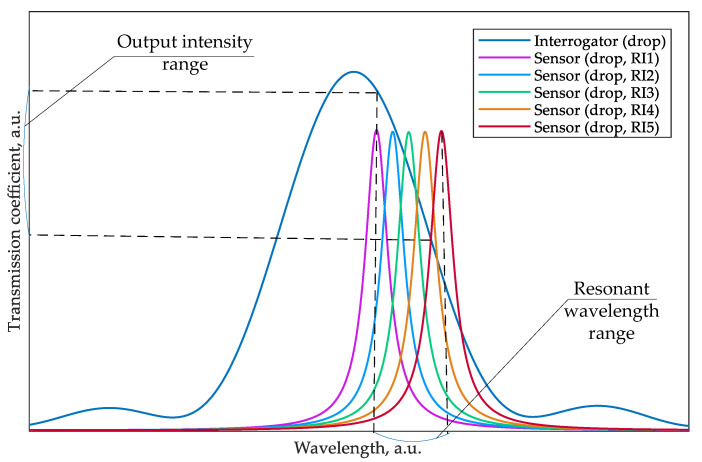
Principle of MRR-based interrogation by intensity.

**Figure 3 sensors-22-09553-f003:**
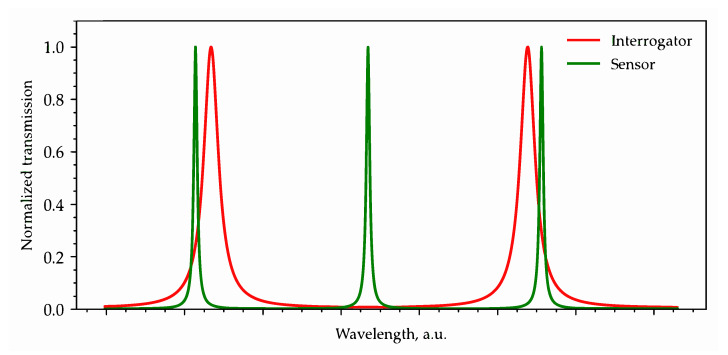
An example of improper overlapping of the interrogator and sensor transmission spectra, which can lead to destructive crosstalk between signals from two sensors’ resonant peaks.

**Figure 4 sensors-22-09553-f004:**
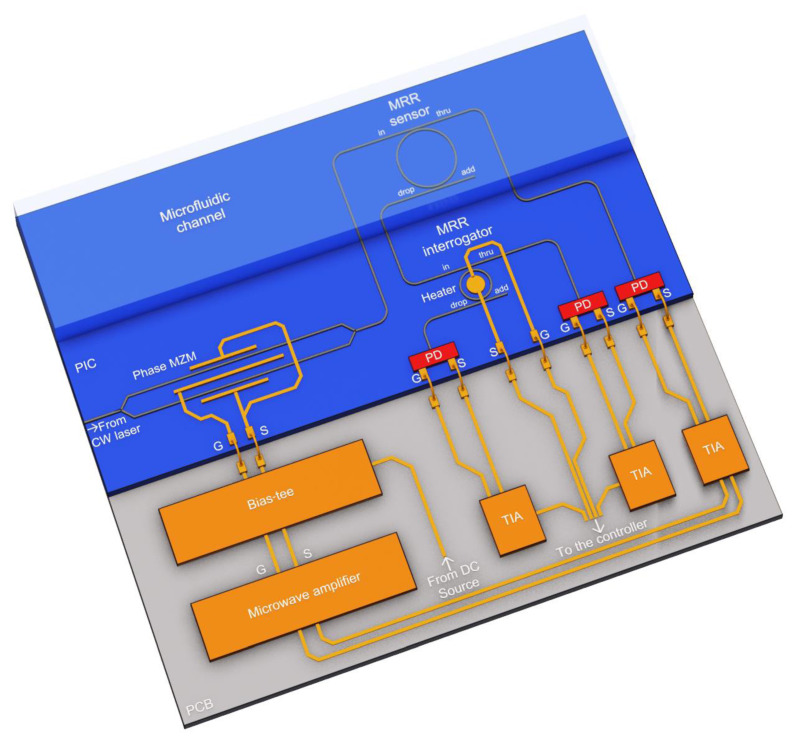
3D design draft of the proposed integrated sensing system with an OEO (not to scale; microwave elements are shown tentatively).

**Figure 5 sensors-22-09553-f005:**
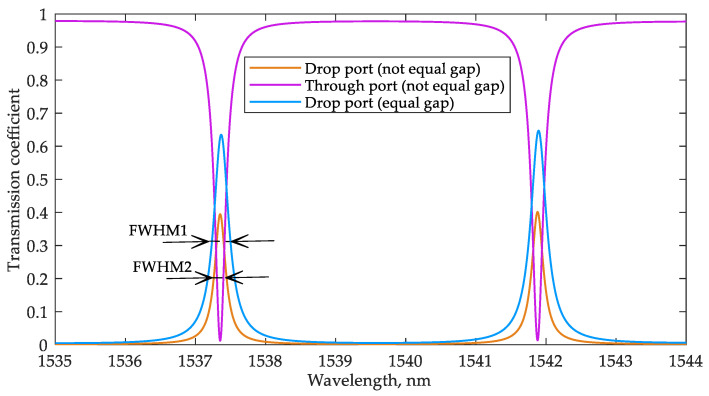
Transmission spectra of the through and drop ports of the sensor MRR.

**Figure 6 sensors-22-09553-f006:**
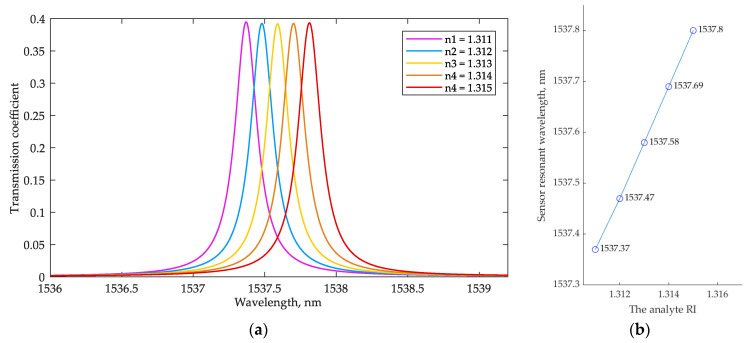
(**a**) The dependence of the sensor MRR transmission spectrum at the drop port on the analyte RI; (**b**) the dependence of the analyzed resonant wavelength of the sensor on the analyte RI.

**Figure 7 sensors-22-09553-f007:**
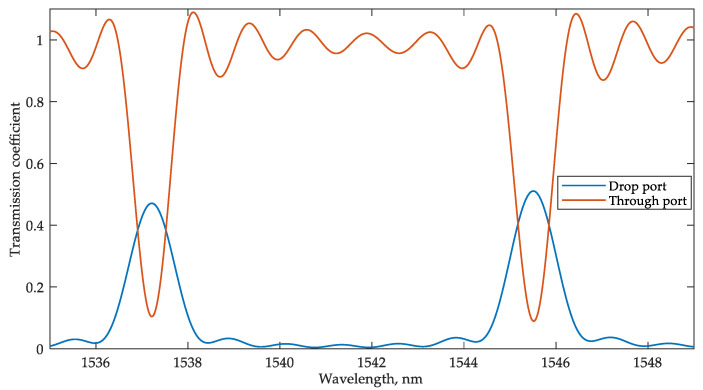
Transmission spectra at the through and drop ports of the interrogator MRR.

**Figure 8 sensors-22-09553-f008:**
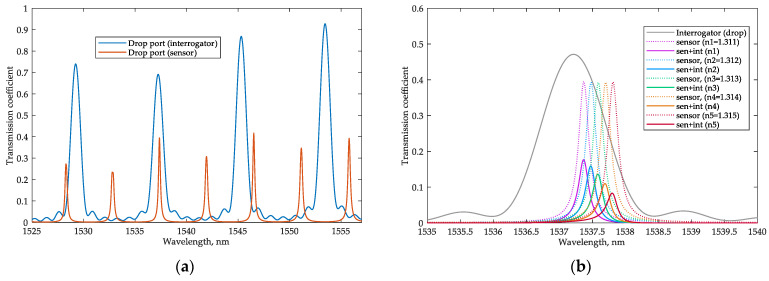
(**a**) Transmission spectra at the drop ports of the sensor and interrogator MRRs; (**b**) Wavelength dependence of the system transmission coefficient for the different analyte RIs.

**Figure 9 sensors-22-09553-f009:**
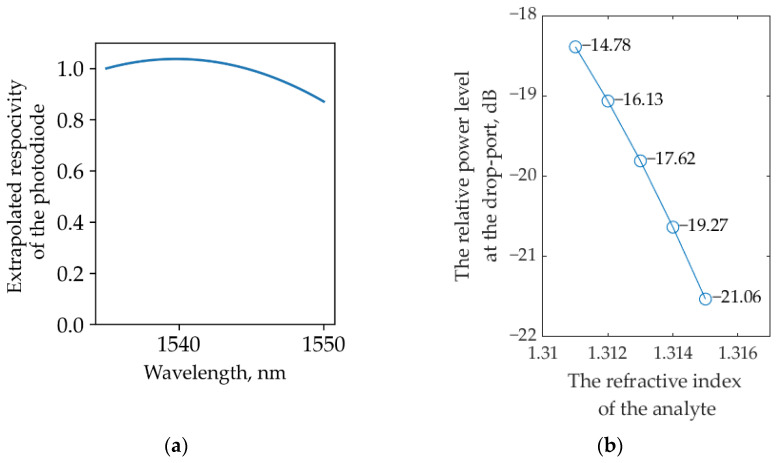
(**a**) Modeled wavelength dependence of the photodiode responsivity; (**b**) Modeled dependence of the relative power at the system output on the analyte RI in the scheme with a broadband source.

**Figure 10 sensors-22-09553-f010:**
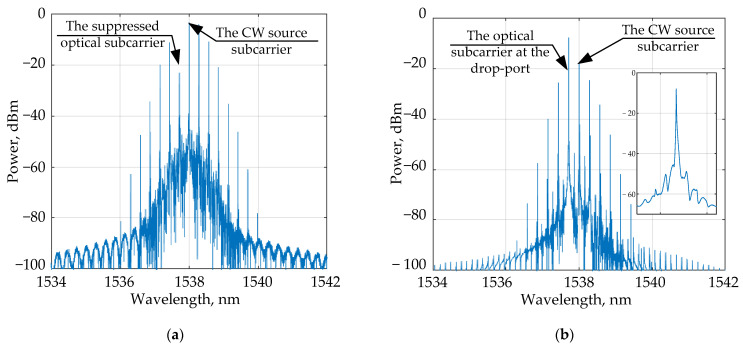
Optical spectrum at the through port (**a**) and the drop port (**b**) of the sensor in the scheme with an OEO. In inset: the magnified spectrum of the optical subcarrier at the drop-port.

**Figure 11 sensors-22-09553-f011:**
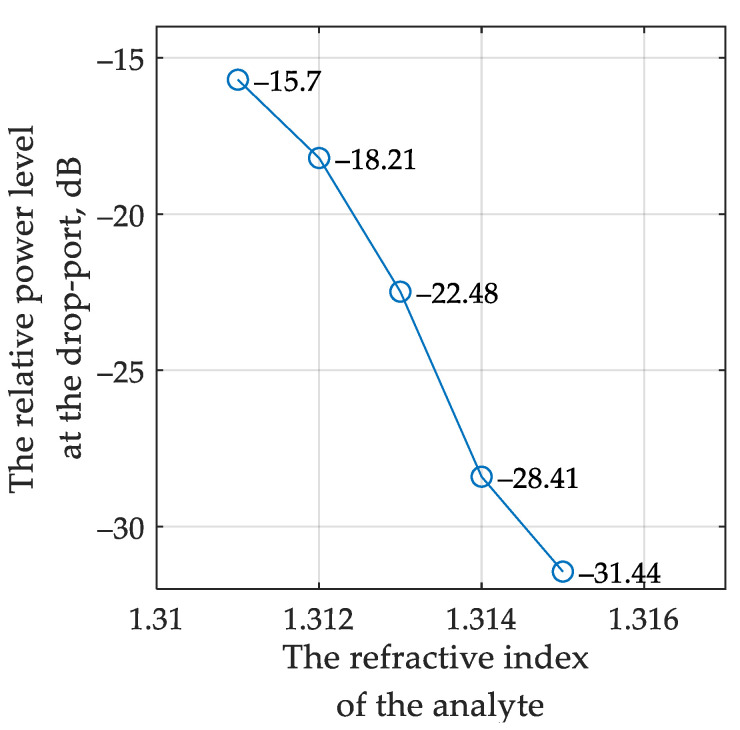
Modeled dependence of the relative power at the system output on the analyte RI in the scheme using an OEO as a source.

**Figure 12 sensors-22-09553-f012:**
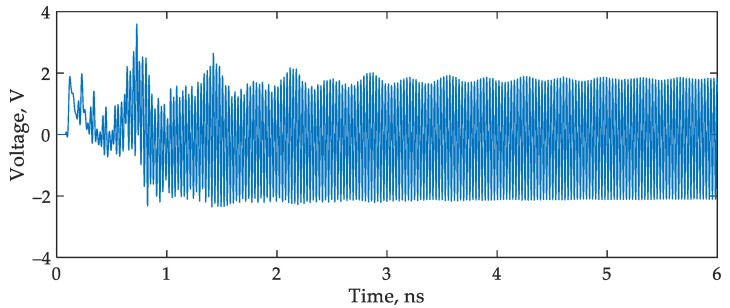
The waveform of the microwave signal at the MZM input.

**Figure 13 sensors-22-09553-f013:**
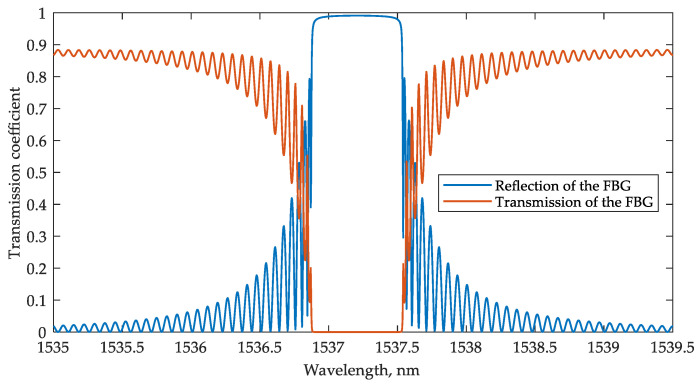
FBG transmission and reflection spectra at the resonant wavelength *λ_B_* = 1537.2 nm.

**Figure 14 sensors-22-09553-f014:**
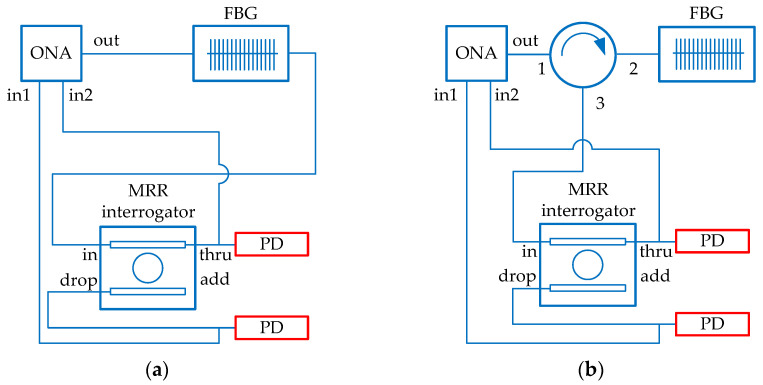
The simulation schemes of FBG interrogation: (**a**) by analyzing the transmitted light; (**b**) by analyzing the reflected light. Numbers 1−3 denote the order of the circulator’s ports.

**Figure 15 sensors-22-09553-f015:**
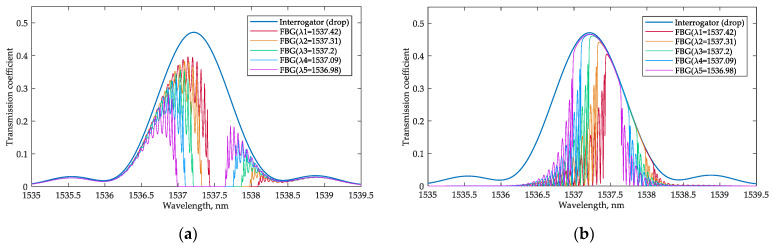
The combined transmission spectra at the interrogator drop port depending on the Bragg wavelength of the FBG: (**a**) for the scheme that analyzes the transmitted light; (**b**) for the scheme that analyzes the reflected light.

**Figure 16 sensors-22-09553-f016:**
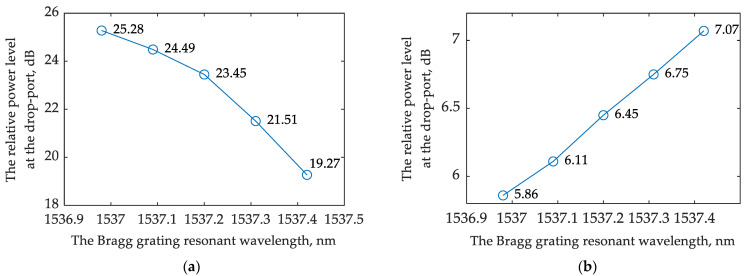
Relative power at the interrogator drop port versus the resonant wavelength of the FBG: (**a**) for the scheme that analyzes the transmitted light; (**b**) for the scheme that analyzes the reflected light.

**Figure 17 sensors-22-09553-f017:**
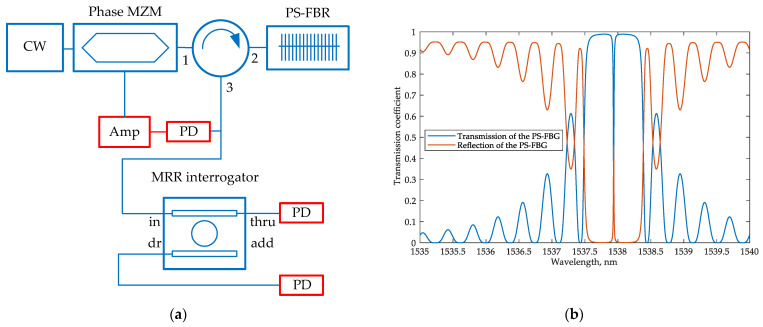
(**a**) PS-FBG interrogation scheme with an OEO; (**b**) PS-FBG transmission and reflection signals spectra at the resonant wavelength *λ_B_* = 1537.2 nm. Numbers 1–3 denote the order of the circulator’s ports.

**Figure 18 sensors-22-09553-f018:**
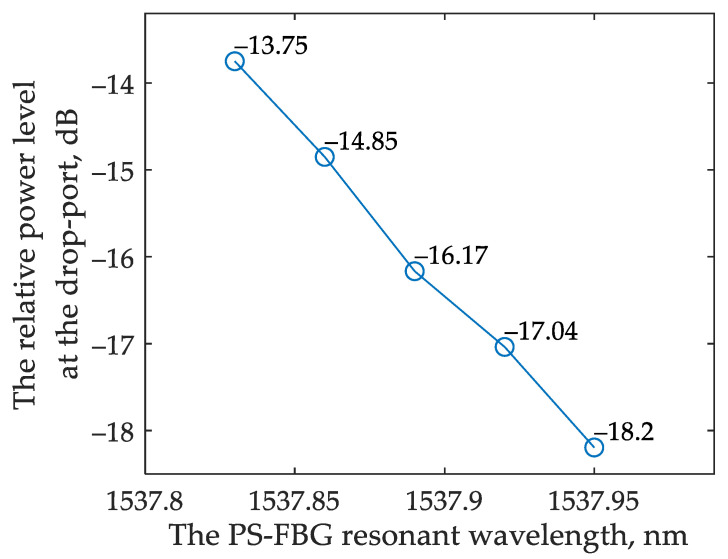
Relative power at the interrogator drop port versus the resonant wavelength of the PS-FBG in the scheme with OEO.

**Figure 19 sensors-22-09553-f019:**
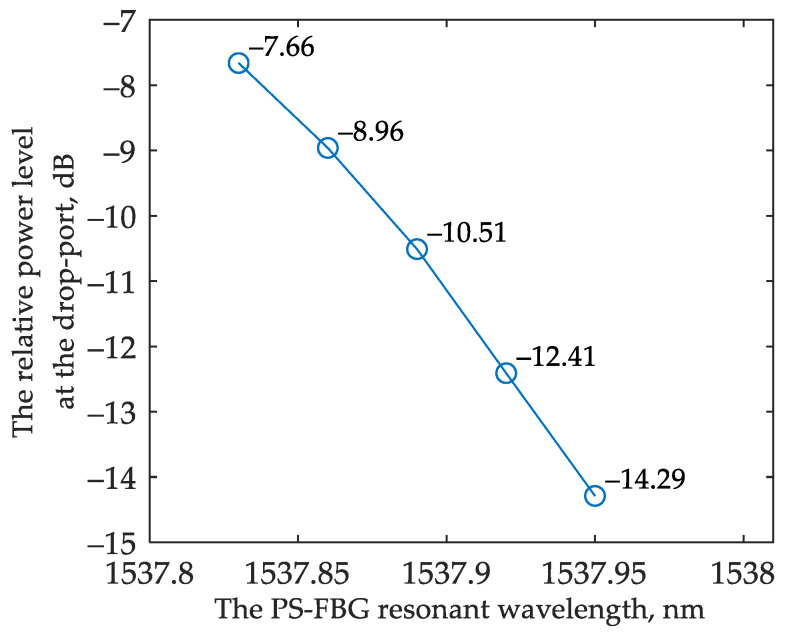
Relative power at the interrogator drop port versus the resonant wavelength of the PS-FBG in the scheme with broadband source.

**Figure 20 sensors-22-09553-f020:**
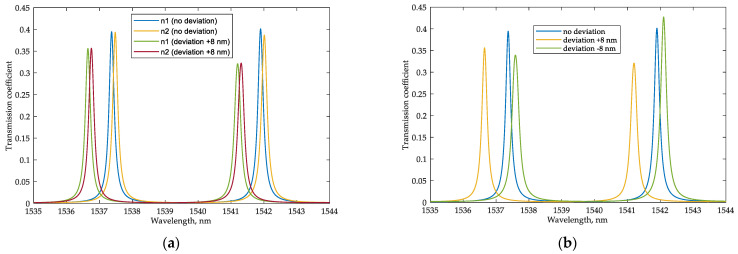
(**a**) Sensor’s transmission spectra for sensing an RI increment of 0.001 (n1 = 1.311, n2 = 1.312) in the case of fabrication deviation of the waveguide width of +8 nm; (**b**) sensor’s transmission spectra for the same analyte RI (n = 1.311) for the fabrication deviation of the waveguide width of ±8 nm.

**Figure 21 sensors-22-09553-f021:**
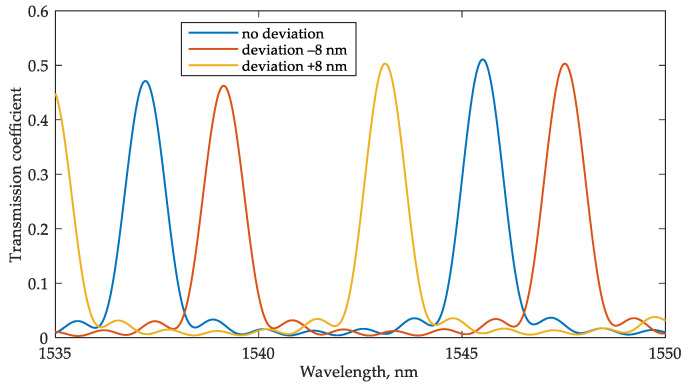
Interrogator’s transmission spectra for the fabrication deviation in the waveguide width of ±8 nm.

**Table 1 sensors-22-09553-t001:** FBG sensor parameters.

Name	Value
Outer cladding diameter	150 µm
Inner core diameter	50 µm
Grating period	0.5 µm
Number of periods	20,000
Effective refractive index	1.5
Periodic variation in the refractive index of the core	10^−3^
Material	SiO_2_

**Table 2 sensors-22-09553-t002:** PS-FBG sensor parameters.

Name	Value
Outer cladding diameter	150 µm
Inner core diameter	50 µm
Grating period	0.51 µm
Phase-shifted aria	0.51 µm
Number of periods	4000
Effective refractive index	1.5
Periodic variation in the refractive index of the core	10^−3^
Material	SiO_2_

## Data Availability

Not applicable.

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
