# Peer review of "Design and Modeling of a Fully Integrated Microring-Based Photonic Sensing System for Liquid Refractometry"

_sensors, 2022, doi:10.3390/s22239553_

Round 1

Reviewer 1 Report (New Reviewer)

The submitted manuscript reports simulation results for an integrated ring resonator-based refractometer sensor without experimental results. Though different possible implementations have been simulated, the manuscript lacks real implementation and experimental results.  In my opinion, the manuscript should not be published in its current form without experimental results.

Other than the lack of experimental results, I have concerns about the simulated designs i.e.

·         Figure 4: Transimpedance amplifiers (TIA), Microwave amplifiers, and Bias-tee can be noticed in the proposed design draft. These components require specialized designs for implementation but authors have not discussed the challenges associated with the implementation of these components.

·         What is the advantage of using the FBG sensor instead of the MRR sensor?

·         Authors have briefly discussed the effect of change in dimensions of the waveguides on the resonance of ring resonators. There was no discussion of variability while discussing the FBG sensor. 

Author Response

The authors thank the Reviewer for the concerns and suggestions. Please, find our replies below.

  • The submitted manuscript reports simulation results for an integrated ring resonator-based refractometer sensor without experimental results. Though different possible implementations have been simulated, the manuscript lacks real implementation and experimental results. In my opinion, the manuscript should not be published in the current form without experimental results.

By our intention, the novelty and impact of this manuscript consist in the design of the sensing system as a whole, where the entire system (both sensor and interrogator) can be implemented on a single PIC. A system-level approach to the design of a sensing device implies solving several technical tasks (such as tailoring the frequency responses of the sensor and interrogator MRRs to each other, overall system performance assessment, enhancing integration level, assessing system tolerance to the fabrication inaccuracy) that we tried to address in our manuscript. To our knowledge, there are no existing works considering such an approach to the design of an entire sensing system, compatible with the standard commercial SOI fabrication process. Moreover, we propose the scheme of a sensing system using OEO with MRR-based interrogation by intensity, which is novel to our knowledge. OEO is usually used in frequency interrogation schemes, where the resonant wavelength shift transduces in a microwave signal. These schemes require high-speed ADC or microwave frequency analyzers to measure the output microwave signal. In contrast, the proposed scheme implements the OEO advantage in sensitivity but does not require complex and expensive microwave electronics for operation.

Indeed, only experimental investigation of the proposed system can provide its detailed verification and practically assess its performance, and we plan it in the near future. However, in the absence of our own fabrication facilities, ordering the PIC fabrication within MPW at external foundries takes at least several months, which, together with custom microfluidic chip assembly and ordering other necessary lab supplies, might cause a rather unpredictable time delay. Therefore, we decided to present our developed design solutions in the current stage, verified by numerical simulations in the widely-accepted software. We still hope our technical proposals might be interesting to the community and gain feedback, and perhaps will be implemented and experimentally tested in a shorter time.

We have also changed the title of the article for a better representation of its content. The new title is “Design and modeling of a fully integrated microring-based photonic sensing system for liquid refractometry”.

  • Figure 4. Transimpedance amplifiers, microwave amplifiers, and Bias-Tee can be noticed in the proposed design draft. These components require specialized design for implementation but authors have not discussed the challenges associated with the implementation of these components.

We certainly agree that the design and development issues of microwave electronic components are important and complex. However, in our opinion, this topic is out of the manuscript's focus. We have supplemented Figure 4 with information that the microwave components are presented tentatively. The same information and options for executing microwave components are given in lines 236-239.

  • What is the advantage of using the FBG sensor instead of the MRR sensor?

FBGs are often used as sensors for monitoring spatially-distributed objects. The relevant information was added in lines 466-470.

  • Authors have briefly discussed the effect of change in dimensions of the waveguides on the resonance of ring resonator. There was no discussion of variability while discussing the FBG sensor.

Of course, inaccuracies in manufacturing the FBG sensors can also affect the final characteristics of the system. Nevertheless, the problems of FBG fabrication are described quite widely in the literature. The main problem may be the change in the Bragg wavelength relative to the calculated one. The described method of adjusting the characteristics of the MRR interrogator makes it possible to adapt the interrogation system to this type of the FBG inscription variability. The relevant information was added in lines 603-607.

Reviewer 2 Report (New Reviewer)

This work reported a refractometric sensing system for liquid analysis with the sensor, and proposed the scheme for its intensity interrogation combined on a single photonic integrated circuit. However, there are some problems in this manuscript. It is recommended to be accepted for the publication in Sensors after a major revision.

1) Subsection is required to make the description clearer in section 2.

2) A comparation of the analytical performance between the developed sensing system and existing systems should be provided to highlight the innovation and advantages of this work.

3)  Please explain why the system sensitivity for analyzing transmitted and reflected light are 3.1 dB/nm and 14.7 dB/nm, and mark in Fig. 17.

4) There are some formatting problems in the references.

5) The quality of figures and tables should be improved (e.g., inconsistently sized fonts).

Author Response

The authors thank the Reviewer for the concerns and suggestions. Please, find our replies below.

1. Subsection is required to make the description clearer in Section 2.

Section 2 was divided into three parts according to the recommendation.

2. A comparison of the analytical performance between the developed sensing system and existing systems should be provided to highlight the innovation and advantages of this work.

The performance characteristics of a sensing system as a whole that we employ in our manuscript, as far as we know, are not given in the existing papers. As a rule, the characteristics of the sensor and the interrogation system are indicated. Therefore, it is not possible to directly evaluate the systemic effect of the proposed solutions in comparison with analogs. The developed MRR-based sensor slightly exceeds the sensitivity of the microring sensors described in the literature. A more detailed comparison of the developed sensor with analogs is added in lines 319-323.

3. Please explain why the system sensitivity for analyzing transmitted and reflected light are 3.1 dB/nm and 14.7 dB/nm, and mark in Fig. 17.

The sensitivity estimate (lines 501-510) was aligned with the methodology described earlier in section 4. Figure 17, that may be misleading, has been deleted; the arithmetic error has been corrected.

4. There are some formatting problems in the references.

Formatting problems have been corrected, and the references list has been revised (Ref. 33 deleted due to duplication; Refs. 51,52 added in accordance with the comment 4 of Reviewer 1).

5. The quality of figures and tables should be improved (e.g. inconsistently sized fonts).

All relevant drawings has been reformatted and brought to a single format.  

Round 2

Reviewer 1 Report (New Reviewer)

The manuscript reports simulation results for an integrated ring resonator-based refractometer sensor. Based on my previous comments authors have changed the title of the article and the new title better represents the research presented in the manuscript. So, the manuscript can be published once the following concerns have been addressed.

·         Figure 3: The plots in the figure are not visible in the revised manuscript. These plots were fine for the initially submitted manuscript so please correct the figure.

·         Figure 6(b): The plot in the figure6(b) are not visible in the revised manuscript. These plots were fine for the initially submitted manuscript so please correct the figure.

·         Figure 9(a): The plots in the figure9(a) are not visible in the revised manuscript. These plots were fine for the initially submitted manuscript so please correct the figure.

·         As authors are discussing the variability for silicon photonics so it would be useful to cite a couple of papers discussing variability analysis and yield prediction for silicon photonics. 

Author Response

The authors thank the Reviewer for the valuable remarks and suggestions. Please, find our replies below.

1) Figure 3: The plots in the figure are not visible in the revised manuscript. These plots were fine for the initially submitted manuscript so please correct the figure.
The problem with figures occurred when exporting the initial Word file to the .pdf format. The pdf version of the manuscript has been corrected.

2) Figure 6(b): The plot in the figure6(b) are not visible in the revised manuscript. These plots were fine for the initially submitted manuscript so please correct the figure.
The pdf version of the manuscript has been corrected.

3) Figure 9(a): The plots in the figure9(a) are not visible in the revised manuscript. These plots were fine for the initially submitted manuscript so please correct the figure.
The pdf version of the manuscript has been corrected.

4) As authors are discussing the variability for silicon photonics so it would be useful to cite a couple of papers discussing variability analysis and yield prediction for silicon photonics.
The manuscript was updated (lines 594-602) according to the comment, and we added two new sources to the references list (50 and 52). 

Reviewer 2 Report (New Reviewer)

The manuscript is recommended to be accepted.

Author Response

Thank you for the appreciation of our manuscript!

This manuscript is a resubmission of an earlier submission. The following is a list of the peer review reports and author responses from that submission.

Round 1

Reviewer 1 Report

The author proposed a refractometer sensor for liquids by using a on-chip racetrack ring resonator as a sensor. The resonant wavelength of the resonator changed with the refractive index of the liquids on the top of the sensor. In this manuscript, the author shows detailed simulation results on the resonator structure, comparison between two interrogation methods, broadband source and OEO source, system performance of cascaded resonator and FBG. However, many papers studied Microring Resonator sensors, already. I have not learned new knowledge from the manuscript. And, it is reluctant to accept publication in this form before the author can provide more solid experimental measurement on the proposed sensor. The experiment results include but not limited to the performance of the proposed resonator, the intensity interrogation method and so on. It is difficult to proceed the manuscript to further consideration before the author supplement the experiment results.

Reviewer 2 Report

The submitted article describes a sensing concept with cascaded microring resonators, with one sensor and one interrogator ring, for use as a liquid refractometric sensor. The main idea is to covert the sensor experienced wavelength shift to an intensity measurement by applying a weighting of the intensity using the through and drop port signals of an overcoupled interrogator microring. Different sensors like a microring and Bragg grating sensor, and an optoelectronic oscillator (OEO) as source signal are presented.

The authors address an important research field with the integration of photonic sensors, however, the article is not convincing because it does not reflect critically on its results. Actually the concept is not that novel as claimed and the presented application scenarios are only superficially examined. The theoretical study is carried out with a commercial simulator tool which would allow for a careful study of the proposed system, however, the analysis is performed with minimal effort on each use case.

Relevant boundary conditions of such a sensing system on a silicon photonic IC are not sufficiently considered: Material dispersion, the periodicity of the resonance peaks and the associated power contributions from other peaks, the linearity of the readout signal, how the spectral overlap of sensor and interrogator ring can be ensured in a real sensing application, how the interrogator filter shape or slope would affect the results etc. The optoelectronic oscillator source approach is interesting but also relies on above considerations.

Statements on integration and conclusions of such a system on a PIC are too simplified. Some reference citations seem random [Refs 10,17], and do not reflect the author's arguments well.

In summary it is not convincingly proven that the proposed concept would work, for instance, in case the sensor or interrogator ring have a dimension offset of only 1nm in heights or widths - which is common in silicon photonics. Hence, I am sorry to say that I cannot recommend the paper to be accepted as a publication.